# Multilevel predictors of anaemia among pregnant women in Ghana: New evidence from the 2019 Ghana Malaria Indicator Survey

**Desmond Klu**[1]*, **Frank Kyei-Arthur**[2], **Margaret Appiah**[2], **Michael Larbi Odame**[3]

**1** Centre for Malaria Research, Institute of Health Research, University of Health and Allied Sciences, Ho, Volta Region, Ghana, **2** Department of Environment and Public Health, University of Environment and Sustainable Development, Somanya, Eastern Region, Ghana, **3** Department of Sustainable Development and Policy, University of Environment and Sustainable Development, Samanya, Eastern Region, Ghana

* klud@uhas.edu.gh

**Data Availability Statement:** The datasets used for this study is openly available and can be accessed via https://dhsprogram.com/.

## Abstract

Anaemia in pregnant women is a major public health concern. A number of multilevel factors have been attributed as contributors to anaemia in pregnancy. The purpose of this study was to examine the multilevel factors predicting anaemia among pregnant women in Ghana. Data for this study were obtained from the 2019 Ghana Malaria Indicator Survey (GMIS) conducted between September 25 and November 24, 2019 in all regions in Ghana. The weighted sample comprised 353 pregnant women aged 15–49 years. Data were analysed with SPSS version 25 using descriptive statistics, Pearson's chi-square test and binary logistic regression modelling. In this study, the outcome variable was anaemia status among pregnant women, while the predictor variables included individual, household, community, and health system level factors. The overall prevalence of anaemia among pregnant women was 28.7%. Of these, 14.5% had mild anaemia, and 13.2% and 1.1% had moderate and severe anaemia, respectively. The results indicate that a higher probability of anaemia in pregnancy is likely to be found among pregnant women less than 35 years (15–24 years, aOR = 3.31; C.I: 1.13–9.73) (25–34 years, aOR = 2.49; C.I:1.06–5.84). A higher likelihood of anaemia was found among pregnant women who did not take SP drug (aOR = 3.70; C.I:1.20–11.43) and also among household heads aged 30–39 years (aOR = 4.51; C.I:1.09–18.71). However, a lower odd of being anaemic was found among pregnant women who had attained secondary or higher education (aOR = 0.19; C.I:0.05–0.76), women in the richest households (aOR = 0.02; C.I:0.00–0.42) and those accessing unimproved drinking water (aOR = 0.37; C.I:0.14–0.95). Furthermore, pregnant women with health insurance coverage had lower probability (aOR = 0.24; C.I: 0.06–0.94) of being anaemic. The results highlight the importance of varying factors at different levels in understanding the prevalence of anaemia among pregnant women. Understanding these factors will play a major contributor to developing strategies and programmes aimed at addressing anaemia among pregnant women.

**Funding:** The authors received no specific funding for this work.

**Competing interests:** The authors have declared that no competing interests exist.

## Background

Anaemia is referred to as a condition in which the level of haemoglobin (Hb) in the body is below normal, which leads to a reduction in the capacity of red blood cells to carry oxygen to body tissues [1]. Pregnant women and children are most affected, even though anaemia affects all population groups. The World Health Organisation (WHO) reports that the global prevalence of anaemia in nonpregnant women, elderly individuals, children, and pregnant women is 29%, 23.9%, 42.6% and 38.2%, respectively [2, 3].

According to the WHO guidelines, anaemia is defined as a blood haemoglobin concentration < 110 g/L for pregnant women [3]. The prevalence of anaemia is an essential health and wellbeing indicator. A study by Balarajan et al. [4] showed that anaemia in pregnancy is more prevalent in developing countries (43%) than in developed countries (9%). In the African region, 9.2 million (46.3%) pregnant women have anaemia, of which 1.5% have severe anaemia [3].

Anaemia may be due to a number of causes, with the most noteworthy contributor being iron deficiency [5]. Approximately 50% of anaemia cases are attributed to iron deficiency [5–7]. Other causes of anaemia include other micronutrient deficiencies (folate, riboflavin and vitamins A and $B_{12}$) and acute and chronic infections (malaria, hookworms infestation, tuberculosis, inherited blood disorders and cancer) [4, 8, 9]. The high prevalence of malaria [10, 11], iron deficiency [11, 12] and micronutrient deficiencies [11, 13, 14] in developing countries, especially in sub-Saharan African countries, could explain why these countries disproportionately bear the greatest burden of anaemia.

Anaemia resulting from iron deficiency adversely affects cognitive and motor development, resulting in low productivity and fatigue [4, 7, 15], and anaemia in pregnancy is often associated with low birth weight and a high risk of maternal and perinatal mortality [16, 17].

In Ghana, 62% of pregnant women have a blood haemoglobin concentration < 110 g/L with an estimated mean blood haemoglobin concentration of 105 g/L [3]. The most recent Ghana Demographic and Health Survey (2022 GDHS) estimated the prevalence of anaemia among pregnant women to be 51% [18]. This figure is higher than the global prevalence of anaemia (38.2%) among pregnant women [3]. Studies in Ghana have found poor knowledge about anaemia [19], maternal underweight, multiple parity [20], and lack of protein in diet during pregnancy [21] as factors predicting anaemia among pregnant women. Although evidence from the literature reveals that anaemia among pregnant women has multifactorial causes that adversely affect maternal and birth outcomes, the multilevel effect among these factors at different levels is understudied and this has implications on the development of comprehensive measures and strategies to address anaemia in pregnancy and ensure safe maternal health. As a result, the multilevel and interaction effects of factors at the individual, household, community, and health system levels on anaemic status of pregnant women are not well known in literature. This study, therefore, examined the effects of multilevel factors that predict anaemia among pregnant women in Ghana. This study will enhance the understanding of how individual, household, community, and health system factors work together to impact anaemia status among pregnant women. In other words, it will help explain the contribution of factors to anaemia prevalence among pregnant women at multiple levels, which will aid in developing strategies and programmes to address anaemia among pregnant women.

## Methods

### Study design and sampling procedure

This study ustilised a cross-sectional secondary data from the 2019 Ghana Malaria Indicator Survey (GMIS), conducted from September 25 to November 24, 2019. The GMIS primary

objectives included collecting information on malaria prevention such as ownership and use of treated mosquito bed nets and assesses coverage of intermittent preventive treatment to protect pregnant women against malaria. Additionally, it aimed to evaluate anaemia levels in pregnant women and children and inquire about malaria prevalence and treatment in Ghana. However, this study specifically focused on anaemia among pregnant women.

A multi-stage sampling procedure was used to recruit respondents for the 2019 GMIS. The rural and urban areas of the 10 regions of Ghana were used as sampling strata for the 2019 GMIS. This resulted in 20 sampling strata. At the first stage, 200 enumeration areas were selected probability proportional using the 2010 Ghana Population and Housing census as the sampling frame. In each selected enumeration area, all households were listed.

In the second stage, 30 households were randomly selected from each of the 200 enumeration areas. In total, 6000 households were randomly selected. In each selected household, women in their reproductive ages (15–49 years) were interviewed.

## Study setting

Ghana, situated in West Africa, shares its borders with Burkina Faso to the North, the Gulf of Guinea to the South, Togo to the East, and Cote d'Ivoire to the West. As of 2021, Ghana has a population of 30.8 million people and is divided into 16 administrative regions [22], with the capital city being Accra of. Over the years, Ghana has implemented various interventions to combat anaemia. These interventions include iron supplementation, food fortification, public education and sensitisation programs, deworming initiatives, as well as management and prevention of parasitic infection during pregnancy [23, 24].

## Data collection

Briefly, the GMIS is a nationally representative survey conducted by the Ghana Statistical Service (GSS), Ministry of Health (MOH) and National Malaria Control Programme of the Ghana Health Service with technical support from the Inner-City Fund (ICF) International through the Demographic and Health Surveys (DHS) Program. Detailed information regarding the scope and methodology of the GMIS have already been published [25]. The data collection for the GMIS occurred in two phases. The first phase involved a household listing exercise, wherein each of the 200 selected enumeration areas were visited. During this phase, information was gathered on structures, as well as the names of household heads and the global positioning system (GPS) coordinates of clusters. The second phase involved interviews with heads of households and all eligible women, including pregnant women aged 15–49 years, who were either usual members of a household or visitors in the selected households.

The total number of women within the reproductive age range of 15–49 years in the 2019 GMIS was 5,181. However, in this study, we limited the analysis to women who reported being pregnant during the survey. Thus, the weighted sample of pregnant women in the 2019 GMIS was 353. Sample weight (v005/1,000,000) was applied in weighting the entire data to correct possible over- and under sampling issues.

## Study variables

**Outcome variable.** Regarding the determination of anaemia levels in pregnant women, a single-use retractable, spring loaded sterile lancet was utilised for the finger prick. Subsequently, a drop of blood from the site was collected in a microcuvette. Haemoglobin analysis was then conducted on-site using a battery-operated portable HaemoCue 201+ analyser, producing results in less than one minute. Anaemia test results were recorded both in the Biomarker Questionnaire and a brochure containing information on the causes and prevention of

anaemia was left with household members. Pregnant women with haemoglobin (Hb) levels below 8 g/dl were advised to seek care at a health facility, accompanied by a referral letter indicating the haemoglobin reading to present to the health worker at the facility. In this survey, pregnant women with Hb < 11g/dL were considered anaemic. Specifically, those with Hb levels 9–10.9g/dL were categorised as having mild anaemia, Hb levels 7–8.9g/dL were classified as moderate anaemia and Hb levels less than 8g/dL were deemed severely anaemic as classified by the World Health Organisation [6].

The outcome variable for this study was anaemia status among pregnant women. Initially, anaemia status among pregnant women was categorised into severe, moderate, mild, and not anaemic. However, it was later recategorised into two groups; pregnant women with severe, moderate, or mild anaemia were coded as "1" indicating being anaemic, while those without anaemia were coded as "0". The rationale behind combining all three anaemia levels into one category is due of the small number of cases with mild (n = 51), moderate (n = 47) and severe (n = 4) anaemia.

**Predictor variables.**   We considered individual, household, community, and health system level factors in this study. The rationale for choosing these factors at different levels is that they may influence the anaemia status differently.

**Individual socio-demographic factors.**   The individual level socio-demographic factors comprised the age of pregnant women (15–24, 25–34, 35–49), educational level (no education, primary, secondary/higher), religion (Catholic, Protestant, Muslim, Pentecostal/Charismatic, no religion), literacy level (illiterate, literate), ethnicity (Akan, Ga/Dangme, Ewe, Mole-Dagbani, other ethnic groups), parity (1–3, 4–6, 7 or more) and trimester of pregnancy (first trimester, 1–12 weeks; second trimester, 13–26 weeks; third trimester, 27 weeks and beyond).

**Household level factors.**   We considered the following household-level factors in the study including sex of household head (male and female), age of household head (20–29,30–39, 40–49, 50–59,60–69, 70+) and household wealth quintile (poorest, poorer, middle, richer, richest). The other variables included household source of drinking water, type of toilet facility and type of cooking fuel used by the household. The measurement and classification of the variable *'household source of drinking water' and the type of toilet facility used* were guided by the WHO/United Nations International Children's Emergency Fund Joint Monitoring Programme for Water Supply, Sanitation and Hygiene (WHO/UNICEF-JMP) classification of drinking water sources. In this study, the variable "household source of drinking water" was categorised into two groups: improved and unimproved sources. Improved sources comprised pipe-borne water inside the dwelling, piped into the dwelling, pipe to yard/plot, piped to the neighbour's house/compound, tube well water, borehole, protected dug well, protected well, protected spring and rainwater collection, bottled water and sachet water. Unimproved sources encompassed unprotected wells, surfaces from spring, unprotected springs, rivers/dam, tanker trucks and carts with small tanks. Similarly, the type of toilet facility was also categorised into improved and unimproved. Improved toilet facilities included flushing to pipe sewers, flushing to septic tanks, flushing to pit latrines, flushing to unknown places, flushing to bio-digesters, ventilated improved pit latrines (VIPs), pit latrines with slabs, pit toilet latrines and composting toilets. Unimproved toilet facilities included flush to somewhere else, pit without slab/open pit, no facility, bush/field and hanging toilet/latrine. Furthermore, the type of household cooking fuel was categorised into the following: liquefied petroleum gas (LPG), charcoal, fuel wood and other cooking fuel (straw/shrub/grass, agricultural crops, and animal dung).

**Community level factors.**   The study also considered community level factors, including place of residence (urban, rural) and ecological zones of residence (coastal zone, middle belt, northern zone). The coastal zone comprises Greater Accra, Volta, Central and Western

regions. The middle belt comprises the Brong Ahafo, Eastern and Ashanti regions while the northern zone included Upper East, Upper West and Northern regions.

**Health system level factors.** In this study, we also examined several health system level factors, including number of antenatal visits (no visit, 1 or more visits), coverage by the National Health Insurance Scheme (Yes, No) and uptake of doses of intermittent preventive treatment of malaria using sulfadoxine pyrimethamine (IPTp-SP) during pregnancy. This was measured using this survey question from the Women's Questionnaire: "*During this pregnancy, did you take SP/Fansidar to keep you from getting malaria?*" The response to the first question was categorised into 1 = yes and 2 = no.

## Statistical analysis

Using version 25 of the Statistical Package for the Social Sciences (SPSS), analyses of the data were performed in three stages. In the first stage, simple descriptive statistics were utilised to describe the outcome and predictor variables. The second stage involved a bivariate analysis or cross-tabulation of all the individual, household, community, and health system level factors against the anaemia status of pregnant women, employing Pearson's chi-square test. In the third stage, using binary logistic regression model, three different multilevel regression models were developed to assess the combined effects of individual, household, community, and health system level factors on anaemia status among pregnant women. Model I analysed the effect of individual and household level factors, while model II considered the effect of community and health system-level factors. The final model examined how individual, household, community and health system factors collectively influenced anaemia status among pregnant women in Ghana. All variables were considered statistically significant at the 95% confidence interval ($p < 0.05$).

## Results

### Anaemia prevalence and severity among pregnant women in Ghana

Fig 1 shows the prevalence of anaemia among pregnant women in Ghana. Out of the 353 pregnant women interviewed, 28.7% of were anaemic, while the remaining 71.3% were not anaemic. Fig 2 shows that out of the 28.7% pregnant women who are anaemic, 1.1% has severe anaemia and 13.2% and 14.4% had moderate and mild anaemia respectively.

### Description predictor variables in the study

Tables 1 and 2 shows the percentage distribution of individual, household and community and health system level factors considered in this study. Among the individual sociodemographic characteristics, 46% of the pregnant women were aged 25–34 years, constituting the highest proportion of pregnant women in any of the age categories, as shown in Table 1. The average age of pregnant women was 28 years. More than half (51.6%) of pregnant women had attained a secondary level of education. With regard to religion, a higher proportion (54.6%) of pregnant women belonged to Pentecostal/Charismatic faith compared to other religious faith. More than half (51.2%) of the pregnant women were literate, while a higher proportion (38.9%) belonged to the Akan ethnic group. Approximately 7 out of 10 pregnant women in Ghana had between 1–3 children, while approximately 5% had 7 or more children. Again, about 61% of pregnant women were in their second trimester (13–26 weeks) pregnancy period.

The results further showed that 77.1% of the pregnant women belonged to male-headed households. The highest proportion (35.4%) of heads of household was between the ages of 30

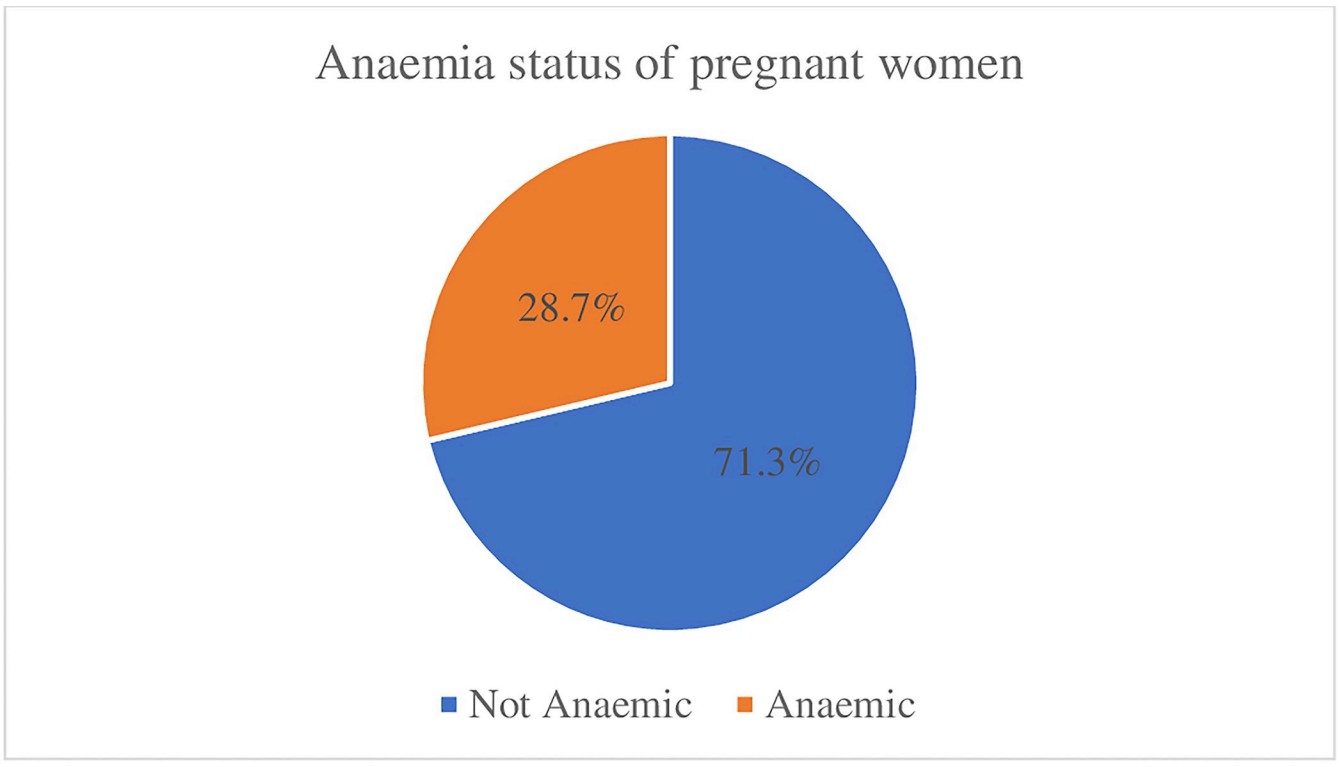

**Fig 1. Anaemia status of pregnant women in Ghana.**

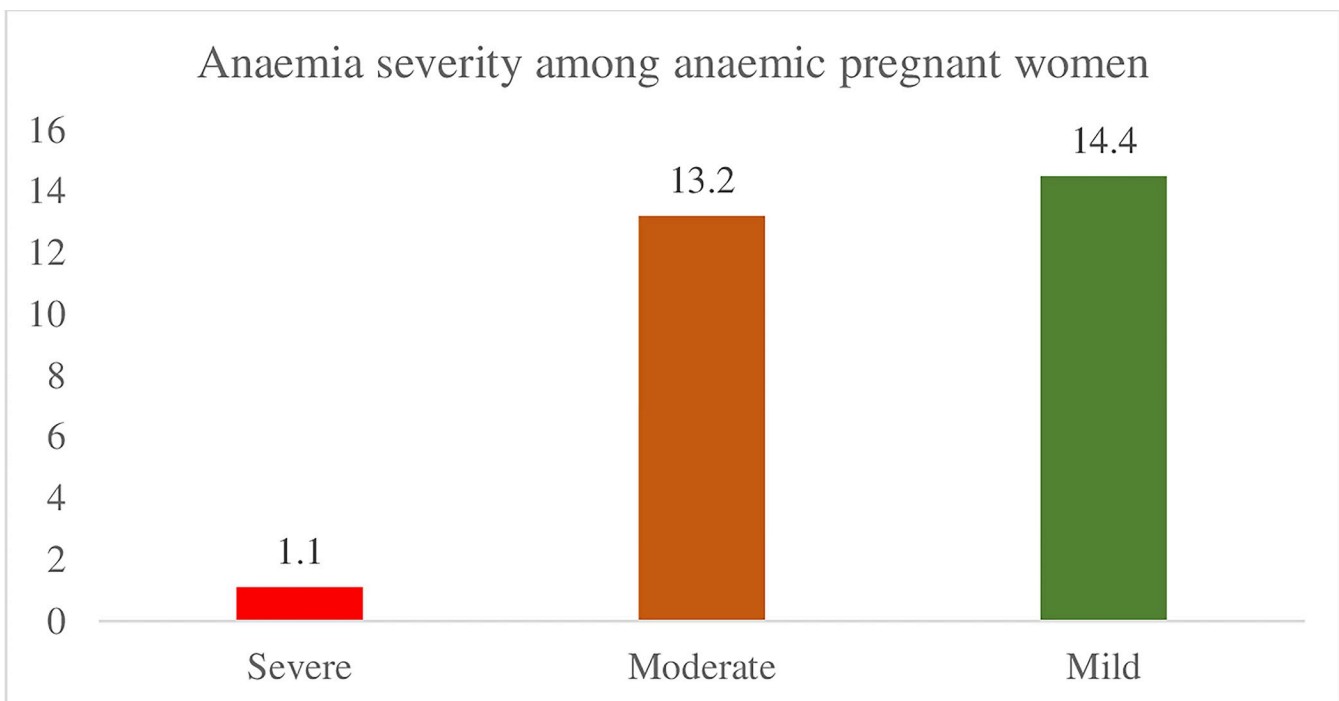

**Fig 2. Anaemia severity among anaemic pregnant women in Ghana.**

**Table 1.  Individual sociodemographic, household characteristics and anaemia status of pregnant women in Ghana.**

| Variables | Weighted Sample n = 353 | % |
|---|---|---|
| *Socio-Demographic Factors* | | |
| **Age** | | |
| 15–24 | 123 | 34.8 |
| 25–34 | 162 | 46.0 |
| 35–49 | 68 | 19.2 |
| **Mean age of pregnant women (± SD)** | **27.92 (±9.11)** | |
| **Education Level** | | |
| No education | 72 | 20.3 |
| Primary | 74 | 21.0 |
| Secondary | 182 | 51.6 |
| Higher | 25 | 7.1 |
| **Religion** | | |
| Catholic | 31 | 8.8 |
| Protestant | 32 | 9.1 |
| Moslem | 81 | 22.8 |
| Pentecostal/Charismatic | 193 | 54.6 |
| No Religion | 16 | 4.6 |
| **Literacy Level** | | |
| Illiterate | 172 | 48.8 |
| Literate | 181 | 51.2 |
| **Ethnicity** | | |
| Akan | 138 | 38.9 |
| Ga/Dangme | 22 | 6.3 |
| Ewe | 67 | 19.1 |
| Mole-Dagbani | 111 | 31.3 |
| Other | 15 | 4.3 |
| **Parity** | | |
| 1–3 | 235 | 66.7 |
| 4–6 | 101 | 28.5 |
| 7+ | 17 | 4.8 |
| **Trimester of Pregnancy** | | |
| First trimester | 110 | 31.0 |
| Second trimester | 214 | 60.5 |
| Third trimester | 30 | 8.4 |
| **Sex of Household Head** | | |
| Male | 272 | 77.1 |
| Female | 81 | 22.9 |
| **Age of Household Head** | | |
| 20–29 | 57 | 16.2 |
| 30–39 | 125 | 35.4 |
| 40–49 | 83 | 23.6 |
| 50–59 | 44 | 12.4 |
| 60–69 | 25 | 7.1 |
| 70+ | 19 | 5.3 |
| **Household Wealth Quintile** | | |
| Poorest | 78 | 22.0 |

*(Continued)*

**Table 1.** (Continued)

| Variables | Weighted Sample n = 353 | % |
|---|---|---|
| Poorer | 74 | 21.0 |
| Middle | 69 | 19.4 |
| Richer | 66 | 18.6 |
| Richest | 67 | 18.9 |
| **Household source of drinking water** | | |
| Improved source of drinking water | 210 | 59.5 |
| Unimproved source of drinking water | 143 | 40.5 |
| **Household type of toilet facility** | | |
| Improved toilet facility | 234 | 66.1 |
| Unimproved toilet facility | 120 | 33.9 |
| **Household type of cooking fuel** | | |
| Liquefied Petroleum Gas (LPG) | 81 | 22.9 |
| Charcoal | 103 | 29.3 |
| Fuel wood | 157 | 44.4 |
| Other cooking fuel | 12 | 3.4 |

Source: Computed from the 2019 Ghana Malaria Indicator Survey (GMIS)

and 39 years. There were more pregnant women (22.0%) in the poorest wealth quintile than in the highest wealth quintile (18.9%). Approximately 6 out of 10 pregnant women belong to households that accessed improved sources of drinking water as well as improved toilet facilities. Additionally, 44.4% of pregnant women belonged to households that used fuel wood as the main type of cooking fuel.

**Table 2. Community and health system characteristics and anaemia status among pregnant women in Ghana.**

| Community level factors | Weighted sample = 353 | Percent |
|---|---|---|
| **Ecological zones of residence** | | |
| Coastal Zone | 136 | 38.6 |
| Middle Belt | 144 | 40.8 |
| Northern Zone | 73 | 20.6 |
| **Type of place of residence** | | |
| Urban | 147 | 41.5 |
| Rural | 207 | 58.5 |
| **Health System level factors** | | |
| **Number of ANC visit** | | |
| No visit | 161 | 45.6 |
| 1+ visit | 193 | 54.5 |
| **NHIS Coverage** | | |
| Covered by NHIS | 91 | 25.8 |
| Not covered by NHIS | 262 | 74.2 |
| **IPTp-SP uptake** | | |
| Did not take SP | 183 | 51.9 |
| Took SP | 170 | 48.1 |

Source: Computed from 2019 Ghana Malaria Indicator Surveys (GMIS)

With regards to community and health system factors, it was found that a higher proportion (40.8%) of pregnant women resided in the middle belt, while more than half (58.5%) of pregnant women resided in rural areas (see Table 3). Approximately 55% of pregnant women attended antenatal care one or more times. This was followed by 45.6% who never visited health facilities for antenatal care. Approximately 7 out 10 indicated they have been covered by the National Health Insurance Scheme compared to 25.8% with no health insurance coverage. Furthermore, a little over half of the pregnant women (51.9%) did not take SP to prevent malaria.

## Association between individual, household, community and health system level factors and anaemic status among pregnant women in Ghana

Bivariate analyses showed association between anaemia and all categories of predictor variables including individual, household, community and the health system level factors. Table 3 shows that the significant factors at the individual level were educational level (p = 0.000), religion (p = 0.004), literacy level (p = 0.000), ethnicity (p = 0.007) and parity (p = 0.000). With regard to household level factors, sex of household head (p = 0.044), wealth index (p = 0.000), type of toilet facility (p = 0.010) and type of cooking fuel (p = 0.011) were significantly associated with anaemia among pregnant women in Ghana at p<0.05. The study also established a significant association between type of place of residence (p = 0.000), number of antenatal care visits (p = 0.000), NHIS coverage (p = 0.027), IPTp-SP uptake (p = 0.000) and anaemic pregnant women at the community and health system levels, respectively.

## Multilevel effect of individual and household factors on anaemia among pregnant women in Ghana

Model I present the results on the collaboration between individual and household factors that influence anaemia among pregnant women in Ghana, as shown in Table 4. The results show that individual socio-demographic factors, such as the age of pregnant women and parity, significantly predicted anaemia among pregnant women in Ghana. In terms of age, the results showed that pregnant women with anaemia aged 15–24 years (aOR = 3.31, CI: 1.13–9.73) and 25–34 years (aOR = 2.49, CI: 1.06–5.84) had higher odds of being anaemic than those aged 35–49 years. Regarding parity, compared to those who had 7 or more children, pregnant women with 1–3 children (aOR = 0.12, CI: 0.03–0.51) had lower odds of being anaemic. At this level, none of the household factors were significant in predicting anaemia among pregnant women when combined with individual level factors.

## Multilevel effect of community factors and health system on anaemia among pregnant women in Ghana

Model II examined the influence of community and health system level factors on anaemia among pregnant women in Ghana (Table 4). Pregnant women residing in urban communities (aOR = 0.48; CI:0.26–0.91) had a lower probability of being anaemic than those living in rural communities. Similarly, pregnant women with no record of antenatal care visits (aOR = 0.00; CI: 0.00–0.03) were less likely to become anaemic than those with one or more visits. Again, pregnant women wo did not take IPTp-SP drug had higher odds (aOR = 3.70; CI: 1.20–11.43) of being anaemic compared to those who took the drug.

**Table 3. Association between individual, household, community and health system level factors and anaemia status among pregnant women in Ghana.**

| Factors | Anaemia Status of Pregnant Women | | |
|---|---|---|---|
| | Not Anaemic n (%) | Anaemic n (%) | P values |
| *Individual Level Factors* | | | |
| *Age group* | | | |
| 15–24 | 96(78.0) | 27(22.0) | 0.092 |
| 25–34 | 108(66.3) | 55(33.7) | |
| 35–49 | 48(70.6) | 20(29.4) | |
| *Educational Level* | | | 0.000*** |
| No Education | 41(57.7) | 30(42.3) | |
| Primary | 44(59.5) | 30(40.5) | |
| Secondary+ | 166(79.8) | 42(20.2) | |
| *Religion* | | | 0.004** |
| Catholic | 27(84.4) | 5(15.6) | |
| Protestant | 26(81.3) | 6(18.8) | |
| Moslem | 45(55.6) | 36(44.4) | |
| Pentecostal/Charismatic | 144(74.6) | 49(25.4) | |
| No Religion | 11(68.8) | 5(31.3) | |
| *Literacy level* | | | 0.000*** |
| Illiterate | 107(61.8) | 66(38.2) | |
| Literate | 145(80.1) | 36(19.9) | |
| *Ethnicity* | | | |
| Akan | 107(77.5) | 31(22.5) | 0.007** |
| Ga/Dangme | 18(81.8) | 4(18.2) | |
| Ewe | 52(76.5) | 16(23.5) | |
| Mole-Dagbani | 68(61.8) | 42(38.2) | |
| Other | 7(46.7) | 8(53.3) | |
| *Parity* | | | 0.000*** |
| 1–3 | 190(80.5) | 46(19.5) | |
| 4–6 | 52(51.5) | 49(48.5) | |
| 7+ | 10(58.8) | 7(41.2) | |
| **Trimester in Pregnancy** | | | |
| First trimester | 84(76.4) | 26(23.6) | 0.286 |
| Second trimester | 149(70.0) | 64(30.0) | |
| Third trimester | 19(63.3) | 11(36.7) | |
| **Household Factors** | | | |
| *Sex of Household Head* | | | |
| Male | 187(68.8) | 85(31.3) | 0.044* |
| Female | 65(80.2) | 16(19.8) | |
| *Age of Household Head* | | | |
| 20–29 | 46(80.7) | 11(19.3) | 0.106 |
| 30–39 | 82(65.6) | 43(34.4) | |
| 40–49 | 55(65.5) | 29(34.5) | |
| 50–59 | 34(79.1) | 9(20.9) | |
| 60–69 | 21(84.0) | 4(16.0) | |
| 70+ | 13(72.2) | 5(27.8) | |

(*Continued*)

**Table 3.** (Continued)

| Factors | Anaemia Status of Pregnant Women | | |
|---|---|---|---|
| *Wealth Index* | | | 0.000*** |
| Poorest | 43(55.1) | 35(44.9) | |
| Poorer | 52(70.3) | 22(29.7) | |
| Middle | 47(68.1) | 22(31.9) | |
| Richer | 51(78.5) | 14(21.5) | |
| Richest | 58(87.9) | 8(12.1) | |
| *Source of drinking water* | | | |
| Improved | 150(71.4) | 60(28.6) | 0.984 |
| Unimproved | 102(71.3) | 41(28.7) | |
| *Type of toilet facility* | | | |
| Improved | 177(75.6) | 57(24.4) | 0.010* |
| Unimproved | 75(62.5) | 45(37.5) | |
| *Type of cooking fuel* | | | 0.011* |
| Liquefied Petroleum Gas | 69(85.2) | 12(14.8) | |
| Charcoal | 73(70.9) | 30(29.1) | |
| Fuel wood | 101(64.7) | 55(35.3) | |
| Other fuel | 8(66.7) | 4(33.3) | |
| **Community level factors** | | | |
| **Ecological zones of residence** | | | |
| Coastal zones | 102(75.0) | 34(25.0) | 0.158 |
| Middle Belt | 104(72.2) | 40(27.8) | |
| Northern zones | 45(62.5) | 27(37.5) | |
| **Type of place of residence** | | | |
| Urban | 120(81.6) | 27(18.4) | 0.000*** |
| Rural | 132(63.8) | 75(36.2) | |
| **Health System level factors** | | | |
| **Number of ANC visit** | | | |
| No ANC visit | 158(98.8) | 2(1.3) | 0.000*** |
| 1+ visit | 94(48.7) | 99(51.3) | |
| **NHIS Coverage** | | | |
| No coverage | 28(56.0) | 22(44.0) | 0.027* |
| Has coverage | 188(71.8) | 74(28.2) | |
| **IPTp-SP uptake** | | | |
| Did not take SP | 165(89.7) | 19(10.3) | 0.000*** |
| Took SP | 87(51.2) | 83(48.8) | |

$p < .05^*, p < .01^{**}, p < .001^{***}$

Source: Computed from 2019 Ghana Malaria Indicator Surveys (GMIS)

## Multilevel effect of individual, household, community and health system factors on anaemia status among pregnant women in Ghana

When all factors (individual, household, community and health system level factors) were considered in Model III, the resulting effect showed that individual factors (educational level of pregnant women), household factors (age of household head, wealth index, and source of drinking water) and health system factors (number of antenatal visits and NHIS coverage) were statistically significant in predicting anaemia among pregnant women in Ghana

**Table 4. Multilevel effects of factors predicting anaemia in pregnancy.**

| Variables | Model I AOR [95%CI | Model II AOR [95% CI] | Model III AOR [95% CI] |
|---|---|---|---|
| *Individual level factors* | | | |
| *Age* | | | |
| 15–24 | **3.31\*(1.13–9.73)** | | 3.01[0.58–15.77] |
| 25–34 | **2.49\*(1.06–5.84)** | | 1.77[0.55–5.74] |
| 35–49 | *Ref* | | Ref |
| *Educational Level* | | | |
| No Education | Ref | | Ref |
| Primary | 1.88(0.81–4.38) | | 1.14[0.31–4.25] |
| Secondary+ | 0.79(0.31–2.02) | | **0.19\*[0.05–0.76]** |
| *Religion* | | | |
| Catholic | 0.21[0.04–1.19] | | 0.20[0.02–2.27] |
| Protestant | 0.95[0.18–4.98] | | 0.41[0.04–4.64] |
| Moslem | 1.72[0.45–6.53] | | 3.47[0.40–30.25] |
| Pentecostal/Charismatic | 1.22[0.31–4.81] | | 2.20[0.28–17.43] |
| No Religion | *Ref* | | *Ref* |
| *Literacy level* | | | |
| Illiterate | *Ref* | | *Ref* |
| Literate | 1.42[0.64–3.15] | | 2.58[0.77–8.67] |
| *Ethnicity* | | | |
| Akan | 0.55[0.13–2.29] | | 1.89[0.18–20.20] |
| Ga/Dangme | 0.33[0.05–1.95] | | 0.34[0.02–5.21] |
| Ewe | 0.41[0.10–1.72] | | 0.84[0.07–10.11] |
| Mole-Dagbani | 0.61[0.17–2.21] | | 0.30[0.03–2.64] |
| Other | *Ref* | | *Ref* |
| *Parity* | | | |
| 1–3 | **0.12\*\*[0.03–0.51]** | | 0.47[0.06–3.62] |
| 4–6 | 0.75[0.21–2.70] | | 1.48[0.24–9.07] |
| 7+ | *Ref* | | *Ref* |
| **Trimester of pregnancy** | | | |
| First trimester | 0.70[0.25–1.94] | | 1.64[0.37–7.35] |
| Second trimester | 0.79[0.30–2.05] | | 1.38[0.39–4.79] |
| Third trimester | *Ref* | | *Ref* |
| *Household level factors* | | | |
| *Sex of Household Head* | | | |
| Male | 1.75[0.81–3.79] | | 1.45[0.44–4.75] |
| Female | Ref | | *Ref* |
| *Age of Household Head* | | | |
| 20–29 | Ref | | *Ref* |
| 30–39 | 1.64[0.64–4.22] | | **4.51\*[1.09–18.71]** |
| 40–49 | 1.49[0.55–4.03] | | 3.14[0.69–14.27] |
| 50–59 | 0.63[0.19–2.08] | | 1.89[0.29–12.50] |
| 60–69 | 0.51[0.12–2.14] | | 1.76[0.16–19.01] |
| 70+ | 1.06[0.27–4.16] | | 2.28[0.32–16.16] |
| *Household Wealth Index* | | | |
| Poorest | Ref | | *Ref* |
| Poorer | 0.94[0.39–2.29] | | 1.23[0.32–4.65] |
| Middle | 0.71[0.24–2.14] | | 0.48[0.09–2.48] |

*(Continued)*

**Table 4.** (Continued)

| Variables | Model I AOR [95%CI] | Model II AOR [95% CI] | Model III AOR [95% CI] |
|---|---|---|---|
| Richer | 0.56[0.16–1.98] | | 0.33[0.05–2.21] |
| Richest | 0.31[0.06–1.58] | | **0.02\*[0.00–0.42]** |
| *Household Source of drinking water* | | | |
| Improved | Ref | | *Ref* |
| Unimproved | 1.44[0.75–2.74] | | **0.37\*[0.14–0.95]** |
| *Household Type of toilet facility* | | | |
| Improved | Ref | | *Ref* |
| Unimproved | 1.27[0.65–2.49] | | 1.18[0.46–3.01] |
| *Household Type of cooking fuel* | | | |
| Liquefied Petroleum Gas | 0.59[0.10–3.66] | | 2.83[0.07–107.88] |
| Charcoal | 0.83[0.15–4.54] | | 0. 79[0.04–16.03] |
| Fuel wood | 0.75[0.15–3.70] | | 0.27[0.02–4.33] |
| Other cooking fuel | Ref | | *Ref* |
| **Community level factors** | | | |
| **Ecological zones of residence** | | | |
| Coastal zones | | *Ref* | |
| Middle Belt | | *1.46[0.71–3.99]* | 1.82[0.66–4.97] |
| Northern zone | | *2.11[0.86–5.16]* | 1.58[0.36–6.98] |
| **Type of place of residence** | | | |
| Urban | | **0.48\*[0.26–0.91]** | 0.58[0.20–1.72] |
| Rural | | *Ref* | Ref |
| **Health System level factors** | | | |
| **Number of ANC visit** | | | |
| No visit | | **0.00\*\*\*[0.00–0.03]** | **0.00\*\*\*[0.00–0.01]** |
| 1+ visits | | *Ref* | *Ref* |
| **NHIS coverage** | | | |
| No coverage | | *Ref* | *Ref* |
| Has coverage | | *0.45[0.19–1.06]* | **0.24\*[0.06–0.94]** |
| **IPTp-SP uptake** | | | |
| Did not take SP | | **3.70\*[1.20–11.43]** | *3.53[0.82–15.17]* |
| Took SP | | *Ref* | *Ref* |

Secondary+ = both secondary and tertiary education; Ref = Reference; AOR: adjusted odds ratio; $p < .05^*$, $p < .01^{**}$, $p < .001^{***}$

Source: Computed from 2019 Ghana Malaria Indicator Surveys (GMIS)

(Table 4). Pregnant women with secondary or higher education (aOR = 0.19; CI: 0.05–0.76) had lower odds of being anaemic than those with no formal education. However, pregnant women with household heads who were relatively older (30–39 years) were more likely (aOR = 4.51, CI: 1.09–18.71) to be anaemic than those with younger household heads (20–29 years). A lower likelihood of being anaemic was found among pregnant women dwelling in the richest households (aOR = 0.02; CI: 0.00–0.42) relative to those who belong to households with the poorest wealth index. Interestingly, pregnant women accessing unimproved sources of drinking water are less likely (aOR = 0.37; CI: 0.14–0.95) to suffer from anaemia compared to those who have access to improved sources of drinking water.

With regard to the effect of health system level factors, pregnant women who did not attend antenatal care had a lower likelihood of being anaemic relative to those who visited health

facilities for antenatal care four or more times. Pregnant women who were covered by the NHIS were less likely (aOR = 0.24, CI: 0.06–0.94) to become anaemic than those who were not covered by the NHIS.

## Discussion

Using the 2019 GMIS data, the study assessed the multilevel effect of individual, household, community and health system level factors on anaemia among pregnant women in Ghana. The results show that anaemia among pregnant women in Ghana is relatively low, with only 2 out of 10 (28.7%) pregnant women being anaemic (HB < 8 g/dl) compared with the 36.5% global anaemia prevalence rate reported by the WHO [3]. Of these anaemic pregnant women, 51 (14.5%) had mild anaemia, 47 (13.2%) had moderate anaemia, and 4 (1.1%) had severe anaemia. Again, the prevalence of anaemia among pregnant women in Ghana is lower than rates reported by studies in Ethiopia (17.8%) [21], Nigeria (17%) [22], (19.2%) [23] and Tanzania (18%) [24]. On the other hand, other studies have found a higher prevalence of anaemia in Morocco (41.37%) [25], Nigeria (58%) [26], Kenya (40%) [27] and Pakistan (51%) [28]. Similarly, relatively high anaemia prevalence rates have been recorded at the facility level, as shown by previous studies in Ghana [19, 20, 26–28]. These facility-based studies reported anaemia prevalence rates among pregnant women ranging between 33% and 57.1%. For instance, a study by Wemakor et al. [19] found anaemia prevalence among pregnant women attending ANC to be 50.8%, with 25% each being mildly and moderately anaemic and less than one percent being severely anaemic. This clearly shows that the prevalence of anaemia among pregnant women using nationally representative data is relatively lower than the prevalence using health facility-based studies. A plausible explanation for the higher prevalence of anaemic in facility-based studies than nationally representative studies is that people who seek treatment for ill-health at a health facility are more likely to undergo laboratory tests, which can aid in the detection of anaemic. Hence, affect the prevalence rate of anaemia in facility-based studies. In addition, the sample sizes for facility-based studies are generally smaller than nationally representative studies, which influence the prevalence rate of a condition since the total sample size serves as the denominator in the calculation of the prevalence rate.

The discrepancy observed in anaemia prevalence among pregnant women in studies could be attributed to different methodological approaches, including the sample and the study settings used by these studies. This may also be a result of advancements in general health care and increases in maternal health care utilisation.

After combining individual level factors with the household, and community and health system level factors, the age of pregnant women, their educational level and parity (individual factors), age of household heads, wealth index, source of drinking water (household factors), type of place of residence (community factors) and number of antenatal visits, insurance coverage and taking sulphadoxine pyrimethamine (health system factor) were significant predictors of anaemia among pregnant women. The study found that pregnant women in Ghana aged less than 35 were more likely to suffer from anaemia during pregnancy compared to those 35 and above. The findings of this study corroborate other studies that also found a high likelihood of anaemia among women less than 35 years [26, 29, 30]. However, it is contrary to other studies that found high anaemia levels among pregnant women aged 35 and above [31–36]. The result also indicates that the likelihood of a pregnant woman being anaemic decreases as her age increases. It is important to note that even though the age of pregnant women is a risk factor for anaemia, some studies found age to be an insignificant predictor of anaemia among pregnant women and women of their reproductive ages [19, 37–43]. Although age alone had no significant effect on anaemia among pregnant women, after interacting the age

of women variable with other variables, the study found age to be a strong significant predictor of anaemia among pregnant women in Ghana.

Similarly, the educational level of pregnant women was also significant in predicting anaemia after interacting it with household, community and health system level factors. The result shows the lower probability of anaemia among pregnant women with a secondary or higher level of education. This finding is similar to previous studies carried out in sub-Saharan Africa, which found that pregnant women with higher education had a significantly lower risk of developing pregnancy-related anaemia [31, 44–48]. This result highlights the importance of formal education in ensuring good maternal health and safe motherhood. Formal education helps pregnant women to be more receptive to modern ideas on pregnancy care practices and adherence to good healthy living to prevent infections during pregnancy [31]. Other studies have observed that a higher educational level of women has been recognised by demographers as a means of decreasing maternal deaths and having better access to information and adherence to dietary and other health recommendations [29].

The findings of this study show a lower risk of anaemia among pregnant women with lower parity (1–3 children) when parity was interacted with household factors (model I). In other words, the higher the number of children (multiple parity) a woman has, the higher the risk of being anaemic during pregnancy. Similar findings have been reported in earlier studies [29, 36, 42, 49, 50]. Some of the reasons for high anaemia among pregnant women with multiparity (5 or more children) are the possibility of repeated drain on the iron reserves with increasing number of children [50], high susceptibility to haemorrhage and maternal depletion syndrome among women with multiple parity [51]. Moreover, women with a high number of children may be sharing available food and resources with other household members, affecting the per capita food intake by pregnant women [52].

Household factors such as age of household head were significant predictors of anaemia status among pregnant women in Ghana when interacted with other individual, household, community and health system factors (Model III). The odds of pregnant women getting anaemia were higher in households with relatively older (30 and above) heads compared to heads of household aged 20–29 years. Other studies did not find any significant relationship between age of household age and anaemia among pregnant women [53]. This result is quite interesting in the sense that older household heads are experienced and expected to know better regarding health issues and complications of pregnancy to prevent anaemia during pregnancy. However, the influence of other individual, household, community and health system factors on the age of the household head may be the explanation for this occurrence.

Another important household factor that was a significant predictor of anaemia among pregnant women was household wealth quintile (Model III). Pregnant women who belong to the richest household wealth quintile were less likely to be anaemic compared to those in the poorest household. This is in line with the findings of other studies in Rwanda [54], India [35], Tanzania [53], Uganda [55] and Ethiopia [56]. This implies that the likelihood of a pregnant women becoming anaemic decreases with increasing household wealth quintile in Ghana. This could probably be explained by the inability of pregnant women in poorer households to afford a balanced diet [57], properly access quality healthcare [58] and practice good hygiene [55].

Another interesting finding of this study was that pregnant women who access unimproved sources of drinking water were less likely to become anaemic compared to those accessing drinking water from improved sources. This finding is in contrast with earlier studies carried out in sub-Saharan Africa, which found a higher risk of anaemia infection among pregnant women who access unimproved sources of drinking water [54, 55, 59]. The reason given is that pregnant women accessing unimproved sources of drinking water are at high risk of

being infected by waterborne diseases and other helminthic infections, such as hookworm, which are considered the most common cause of anaemia in poor sanitary conditions [47]. Other studies found no significant relationship between household sources of drinking water and anaemia in pregnant women [47, 60, 61]. This finding signified a complete departure from the findings of other studies, therefore highlighting the importance of the multilevel effect of other factors on the association between household source of drinking water and anaemia status of pregnancy.

Community level factors such as type of place of residence were significant predictors of anaemia in Ghana. The results also revealed that pregnant women who dwell in urban communities had lower odds of being anaemic than their rural counterparts. This corroborated finding is similar to the findings of earlier studies that also found low anaemia prevalence among pregnant women in urban areas [36, 38, 45, 55]. The plausible reasons are that pregnant women in rural communities may lack information about adequate nutrition, and there is inadequate access to health care centres in rural areas. However, studies performed in East Africa [47, 62] found a lower risk of anaemia infection among pregnant women living in rural communities than among those in urban areas. The probable explanation given was that, women living in rural communities often have access and use herbal and other iron-containing foods that lead to a decrease in the risk of nutritional anaemia.

The number of ANC visits by pregnant women and NHIS coverage were significant health system level factors predicting anaemia among pregnant women in Ghana. Surprisingly, the study findings show a lower risk of anaemia infection among pregnant women who never visit health facilities for antenatal care. This result may be due to the influence of other variables in the model. This is not consistent with previous studies, which found high anaemia among pregnant women with fewer than 4 antenatal visits [46, 63]. A plausible explanation could be the use of alternative or herbal and iron-rich food by pregnant women who are not going for antenatal care [46]. Moreover, infrequent antenatal care visits by pregnant women may have resulted in the delay or inability to access and take iron-folic supplementation and other medications [43]. Consistent with other findings, pregnant women with NHIS coverage were less likely to become anaemic compared to those with no coverage. Similar findings have been reported by other studies, which found low anaemia levels among persons with high health insurance coverage [64]. This highlights the importance of health insurance intervention in improving maternal health outcomes.

## Limitation

The main limitation of the study is that since this is a cross-sectional study, we are unable to establish causality among the predictor and outcome variables. Another limitation was the omission of other important factors that affect anaemia in pregnancy such as nutritional status of pregnant women, cultural practices, and further laboratory testing to confirm anaemic status due to the use of secondary data. However, this study is nationally representative, and it provides insight and nuances into the varying effects of individual, household, community and health system level factors that predict anaemia outcomes among pregnant women in Ghana.

## Conclusion

This study examined the multilevel effect of factors (individual, household, community, and health system) on anaemia status among pregnant women in Ghana. This study highlights and affirms the importance of varying factors at different levels as far as anaemia among pregnant women is concerned. After the inclusion of all factors in the model, it was established that

pregnant women aged 20–35 years (individual) and those whose household heads (household) are aged 30–39 years have greater odds of being anaemic during pregnancy. On the other hand, pregnant women with 1–3 children, those who attained a secondary/higher education level (individual), belonging to the richest household wealth quintile, access to unimproved sources of drinking water (household), reside in urban communities (community), have no ANC visitation, taking IPTp-SP drug and have NHIS coverage (health system) have a lower risk of having anaemia in Ghana.

These findings suggest that when developing strategies and programs to eradicate anaemia in pregnant women, factors should not be considered in isolation. These significant individual, household, community and health system level factors should be considered when developing interventions to improve and strengthen anaemia prevention strategies among pregnant women in Ghana.

## Author Contributions

**Conceptualization:** Desmond Klu.

**Data curation:** Desmond Klu.

**Formal analysis:** Desmond Klu, Frank Kyei-Arthur.

**Methodology:** Desmond Klu, Frank Kyei-Arthur, Margaret Appiah.

**Validation:** Desmond Klu.

**Writing – original draft:** Desmond Klu.

**Writing – review & editing:** Frank Kyei-Arthur, Margaret Appiah, Michael Larbi Odame.

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
