## [Decision Letter · Decision Letter 0]

4 Apr 2024

PGPH-D-23-00541

Multilevel predictors of anaemia among pregnant women in Ghana: New Evidence from the 2019 Ghana Malaria Indicator Survey

Dear Dr. Klu,

Thank you for submitting your manuscript to PLOS Global Public Health. After careful consideration, we feel that it has merit but does not fully meet PLOS Global Public Health’s publication criteria as it currently stands. Therefore, we invite you to submit a revised version of the manuscript that addresses the points raised during the review process.

Please note that we have only been able to secure a single reviewer to assess your manuscript. We are issuing a decision on your manuscript at this point to prevent further delays in the evaluation of your manuscript. Please be aware that the editor who handles your revised manuscript might find it necessary to invite additional reviewers to assess this work once the revised manuscript is submitted. However, we will aim to proceed on the basis of this single review if possible. 

We look forward to receiving your revised manuscript.

Kind regards,

Annesha Sil, Ph.D.

Staff Editor, PLOS

Journal Requirements:

2. Please provide separate figure files in .tif or .eps format only and remove any figures embedded in your manuscript file. Please also ensure all files are under our size limit of 10MB.

- https://doi.org/10.1186/s12884-022-04586-2

- http://dx.doi.org/10.1097/GRH.0000000000000063

- https://doi.org/10.1186/s12887-023-03919-0

In your revision ensure you cite all your sources (including your own works), and quote or rephrase any duplicated text outside the methods section. Further consideration is dependent on these concerns being addressed.

4. In the online submission form, you indicated that "Data will be made available upon request". All PLOS journals now require all data underlying the findings described in their manuscript to be freely available to other researchers, either 1. In a public repository, 2. Within the manuscript itself, or 3. Uploaded as supplementary information.

Additional Editor Comments (if provided):

Reviewers' comments:

Reviewer's Responses to Questions

**Comments to the Author**

1. Does this manuscript meet PLOS Global Public Health’s publication criteria? Is the manuscript technically sound, and do the data support the conclusions? The manuscript must describe methodologically and ethically rigorous research with conclusions that are appropriately drawn based on the data presented.

Reviewer #1: Yes

2. Has the statistical analysis been performed appropriately and rigorously?

Reviewer #1: Yes

3. Have the authors made all data underlying the findings in their manuscript fully available (please refer to the Data Availability Statement at the start of the manuscript PDF file)?

Reviewer #1: Yes

4. Is the manuscript presented in an intelligible fashion and written in standard English?

Reviewer #1: Yes

5. Review Comments to the Author

Reviewer #1: Multilevel predictors of anaemia among pregnant women in Ghana: New Evidence

from the 2019 Ghana Malaria Indicator Survey

Title

The title is well-defined with much precision

Abstract

The section is well structured and written. However, ‘factors’ is erroneously written as ‘factrs’ at the results section. Authors are advised to carefully read the section and correct all punctuations as expected.

Background

It’s interesting to note how author brings to bear the common perceived need for this study. However, the problem statement is not clearly established; hence, authors are advised to clearly establish the claim for this study since a number of studies on the subject matter have been written in Ghana, with some focusing on a myriad of multilevel effects of the subject matter.

Methods

The section and analyses make the paper a well-grounded one and as a serious academic paper. The section is succinct and on point. However, matters and information on sampling and research design are not enough.

Authors are to stick to a particular writing style since both the British and American styles are combined in the manuscript ‘anaemia’ ‘sensitization.’ Either solely British or American is advised.

The last sentence of the Data collection sub-section uses ‘resting’ instead of ‘testing’ Authors are advised to make necessary correction.

Discussion

The discussion section is nicely structured; however, a few portions are poorly cited. In the first paragraph of the section, authors make claim on the comparison of anaemia prevalence in the study area and the global rate, where referenced item 19 is wrongly cited. Authors are advised to make amends.

References

The section is nicely carved out

6. PLOS authors have the option to publish the peer review history of their article (what does this mean?). If published, this will include your full peer review and any attached files.

**Do you want your identity to be public for this peer review?** For information about this choice, including consent withdrawal, please see our Privacy Policy.

Reviewer #1: **Yes: **Jones Asafo Akowuah

---

## [Decision Letter · Decision Letter 1]

14 Aug 2024

Multilevel predictors of anaemia among pregnant women in Ghana: New Evidence from the 2019 Ghana Malaria Indicator Survey

PGPH-D-23-00541R1

Dear Dr. Desmond Klu,

We are pleased to inform you that your manuscript 'Multilevel predictors of anaemia among pregnant women in Ghana: New Evidence from the 2019 Ghana Malaria Indicator Survey' has been provisionally accepted for publication in PLOS Global Public Health.

Best regards,

Wanjiku N Gichohi-Wainaina, Ph.D

Academic Editor

Reviewer Comments (if any, and for reference):

Reviewer's Responses to Questions

**Comments to the Author**

1. If the authors have adequately addressed your comments raised in a previous round of review and you feel that this manuscript is now acceptable for publication, you may indicate that here to bypass the “Comments to the Author” section, enter your conflict of interest statement in the “Confidential to Editor” section, and submit your "Accept" recommendation.

Reviewer #1: All comments have been addressed

2. Does this manuscript meet PLOS Global Public Health’s publication criteria? Is the manuscript technically sound, and do the data support the conclusions? The manuscript must describe methodologically and ethically rigorous research with conclusions that are appropriately drawn based on the data presented.

Reviewer #1: Yes

3. Has the statistical analysis been performed appropriately and rigorously?

Reviewer #1: Yes

4. Have the authors made all data underlying the findings in their manuscript fully available (please refer to the Data Availability Statement at the start of the manuscript PDF file)?

Reviewer #1: Yes

5. Is the manuscript presented in an intelligible fashion and written in standard English?

Reviewer #1: Yes

6. Review Comments to the Author

Reviewer #1: I am really satisfied with the state of the manuscript

7. PLOS authors have the option to publish the peer review history of their article (what does this mean?). If published, this will include your full peer review and any attached files.

**Do you want your identity to be public for this peer review?** For information about this choice, including consent withdrawal, please see our Privacy Policy.

Reviewer #1: **Yes: **Akowuah Jones Asafo
